# Driven-dissipative quantum battery with nonequilibrium reservoirs

Zhihai Wang[1,2], Hongwei Yu[1] and Jin Wang[3]⋆

**1** State Key Laboratory of Electroanalytical Chemistry, Changchun Institute of Applied Chemistry, Chinese Academy of Sciences, Changchun 130022, China
**2** Center for Quantum Sciences and School of Physics, Northeast Normal University, Changchun 130024, China
**3** Department of Chemistry and Department of Physics and Astronomy, State University of New York at Stony Brook, New York 11794, USA
⋆ jin.wang.1@stonybrook.edu

June 29, 2023

## Abstract

We investigate a quantum battery system which yields both of the driven and dissipation. Furthermore, the coupled two-level charger and battery are immersed in nonequilbrium boson or fermion reservoirs. We consider the change of the energy spectrum induced by the external driving to the charger by going beyond the secular approximation and obtain the Redfield master equation. When the charger and the battery possess the same transition frequency and the charger is driven in resonance, a bistability can emerge with the closure of the Liouvillian gap. As a result, the efficiency of the battery depends on the initial state of the charger-battery system, and certain types of entangled initial states can enhance the efficiency. In the non-resonance driving regime, the efficiency of the quantum battery can be optimized by the compensation mechanism for both the boson and fermion reservoirs. Our investigation is helpful to the design and optimization of quantum battery in the nonequlibrium open system.

# 1 Introduction

The stage is yours. Write your article here. The bulk of the paper should be clearly divided into sections with short descriptive titles, including an introduction and a conclusion. A central issue in quantum technology is to explore how to utilize the quantum resource to accomplish various tasks which can not be realized in the classical counterpart. One of such tasks originates from the energy storage, which coins the word "quantum battery". A typical quantum battery device is composed of a charger which supplies the energy to the battery, and a battery which is used to store and extract energy.

Ever since the concept of quantum battery was proposed by Alicke and Fannes [1], several quantum battery models have been put forward in different physical systems, for example, the spin or resonator chain model [2–7, 7–15], the Tavis-Cummings and Dicke model in quantum optics [16–21], Rydberg atom system [22]. In some of these systems, the Floquent technology has been used to enhance the performance of the quantum battery [23–25]. Since a quantum system is inevitable to couple to the reservoirs, the quantum battery which is subject to the open system is now also evoking significant interests [26–36].

The simplest open system for quantum battery is probably two coupled two-level-system setup, which is subject to the external environments. One of them is driven by a classical field and can serve as the charger and the other one can serve as the battery [28]. In this paper, we further couple the charger and battery with two independent reservoirs that can exchange energy (for boson reservoirs) or particles (for fermion reservoirs) with the system. We emphasize that our model is established by a non-equilibrium setup, where the temperature difference for boson reservoirs or chemical potential difference for fermion reservoirs supplies the non-equilibrium for the system.

We adopt a quantum master equation approach under the Born-Markovian approximation. Beyond the Lindblad master equation [37], we derive the Redfield master equation [38, 39], which is widely applied in the study of quantum transport [40, 41] and photosynthetic reactions [42–44]. To obtain the Redfield master equation, we work in the eigen states representation by regarding the driving field as part of the open system, since it actually changes the energy spectrum of the charger-battery system. This is dramatically different from that in the traditional treatment [28], where the driving is treated as an effective reservoir and is introduced phenomenologically. Furthermore, by taking the non-equilibrium into consideration, which was shown to induce the steady state coherence and entanglement [45–55], one can also go beyond the secular approximation, and find out whether the steady state entanglement is beneficial to increase the efficiency of the quantum battery.

One of the surprising results based on our Redfield master equation approach is the emergence of bistability when the charger and the battery are identical and the charger is driven in resonance. Such bistability is associated with the Liouvillian gap closure [56–58], and the efficiency of the quantum battery when the charging time becomes very long depends on its initial state. As for the traditional Lindblad master equation treatment, it yields a unique steady state, so that the efficiency is initial state independent.

In the above mentioned identical and resonant driving case, certain types of initial entanglements can lead to the steady state to be entangled and beneficial for the performance of the quantum battery. The efficiency can reach its maximum value when the system is immersed in the equilibrium boson or fermion reservoirs in this case. On the contrary, when the tran-

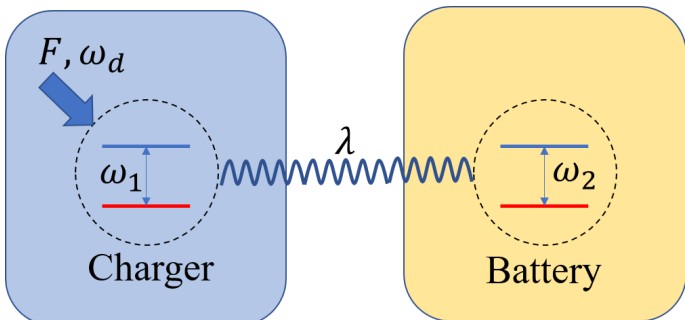

Figure 1: Schematic diagram of the quantum battery model under consideration. The charger and the battery are both two-level systems with transition frequencies $\omega_1$ and $\omega_2$, respectively. They are immersed in their individual reservoir and coupled to each other with coupling strength $\lambda$. The charger is driven by an external classical field with strength $F$ and frequency $\omega_d$.

sition frequencies of the charger and the battery differ from each other, the efficiency of the battery is enhanced more than 90% by setting the frequency of the battery larger than that of the charger and coupling it to the boson reservoir with higher temperature. For the case of fermion reservoirs, the efficiency is dominantly determined by the charger-battery detuning. Whether red or blue detuning is beneficial to enlarge the efficiency depends on the sign of the average chemical potential. In this sense, the efficiency diagram as a function of detuning, nonequilibrium as well as the driving strength provides important guidelines for enhancing the efficiency of our simple quantum battery setup.

   The rest of the paper is organized as follows. In Sec. 2, we illustrate our model and derive the Redfield master equation. In Sec. 3, we discuss the bistability emergence and the steady state entanglement in our system. The efficiency of quantum battery is investigated in resonant and non-resonant driving in Sec. 4 and Sec. 5, respectively. At last, we give a short conclusion in Sec. 6.

## 2   Model and master equation

As schematically shown in Fig. 1, the quantum battery system under our consideration is composed of a charger and a battery, which are both considered as two-level systems. The charger is driven by a classical field, the charger and the battery are coupled to their individual reservoirs following either bosonic or fermionic statistics. In the rotating frame with respective to the frequency $\omega_d$ of the driving field, the Hamiltonian of the whole system including reservoirs reads $H = H_s + H_B + V$. Here, the Hamiltonian for the charger-battery system is [28] ($\hbar = k_B = 1$ in what follows)

$$
\begin{aligned}
H_s &= \frac{\Delta_1}{2}\sigma_z^{(1)} + \frac{\Delta_2}{2}\sigma_z^{(2)} + \lambda\left(\sigma_+^{(1)}\sigma_-^{(2)} + \sigma_-^{(1)}\sigma_+^{(2)}\right) \\
&\quad + \frac{F}{2}\left(\sigma_+^{(1)} + \sigma_-^{(1)}\right).
\end{aligned}
\tag{1}
$$

Here, $\sigma_m^{(i)}(m = z, +, -)$ is the Pauli operator for the $i$th two-level system with transition frequency $\omega_i$, and the notation 1 and 2 represents the charger and battery, respectively. $\Delta_i = \omega_i - \omega_d$ is the detuning between the charger/battery and the driving field, $F$ is the driving strength and $\lambda$ is the coupling strength between the battery and the charger.

The Hamiltonian of the reservoirs reads

$$H_B = \sum_k (\omega_{bk} - \omega_d) b_k^\dagger b_k + \sum_k (\omega_{ck} - \omega_d) c_k^\dagger c_k, \tag{2}$$

where $b_k$ $(b_k^\dagger)$ and $c_k$ $(c_k^\dagger)$ are the annihilation (creation) operators for the $k$th mode with frequencies $\omega_{bk}$ and $\omega_{ck}$ in the reservoirs in contact with the charger and the battery, respectively. The Hamiltonian for the system-reservoir coupling can be expressed as

$$V = \sum_k g_k \sigma_x^{(1)} \left( b_k^\dagger + b_k \right) + \sum_k f_k \sigma_x^{(2)} \left( c_k^\dagger + c_k \right), \tag{3}$$

where $\sigma_x^{(i)} = \sigma_+^{(i)} + \sigma_-^{(i)}$ and $g_k$ $(f_k)$ denotes the coupling strength between the charger (battery) and the $k$th mode in its contacting reservoir. One should note that we here keep both of the formal rotating wave terms (i.e., $\sigma_+^{(1)} b_k + \sigma_-^{(1)} b_k^\dagger$ and $\sigma_+^{(2)} c_k + \sigma_-^{(2)} c_k^\dagger$) and counter-rotating wave terms (i.e., $\sigma_+^{(1)} b_k^\dagger + \sigma_-^{(1)} b_k$ and $\sigma_+^{(2)} c_k^\dagger + \sigma_-^{(2)} c_k$) in the system-reservoir coupling interaction Hamiltonian.

Under the significant approximation and simplification, the conventional Markovian master equation for the charger-battery system can be written as

$$\frac{d}{dt}\rho = -i[H_s, \rho] + \sum_{i=1}^{2} J_i(\omega_i) N_i(\omega_i) D_{\sigma_+^{(i)}}[\rho] + \sum_{i=1}^{2} J_i(\omega_i) \mathcal{N}_i(\omega) D_{\sigma_-^{(i)}}[\rho], \tag{4}$$

where $D_A[\rho] = 2A\rho A^\dagger - A^\dagger A\rho - \rho A^\dagger A$, $J_1(\omega) = \pi \sum_k g_k^2 \delta(\omega - \omega_1)$ and $J_2(\omega) = \pi \sum_k f_k^2 \delta(\omega - \omega_2)$ are respectively the spectra of the two reservoirs. One should note that the formal counter-rotating wave terms in Eq. (3) play no roles in the above master equation. For the boson reservoirs, the average particle number on frequency $\omega$ in the $i$th reservoir is $N_i(\omega) = [\exp(\omega/T_i) - 1]^{-1}$ with $T_i$ being the temperature of the $i$th reservoir and $\mathcal{N}_i(\omega) = N_i(\omega) + 1$. For the fermion reservoirs, it becomes $N_i(\omega) = \{\exp[(\omega - \mu_i)/T_i] + 1\}^{-1}$ with $\mu_i$ being the chemical potential of the $i$th reservoir and $\mathcal{N}_i(\omega) = 1 - N_i(\omega)$.

For the above approximated master equation, the effects of both the external driving and the charger-battery coupling to the energy spectrum are neglected. As a result, the master equation yields a Lindblad form. In what follows, we will derive the master equations by taking into account of both of the above two effects and obtain the Redfield master equations when the classical driving is in and off resonance, respectively.

## 2.1 Resonant driving

We first consider that the charger and the battery are identical and the charger is driven in resonance, that is, $\Delta_1 = \Delta_2 = 0$. Then, the Hamiltonian of charger-battery system becomes

$$H_s = \lambda[\sigma_+^{(1)} \sigma_-^{(2)} + \sigma_-^{(1)} \sigma_+^{(2)}] + \frac{F}{2}[\sigma_+^{(1)} + \sigma_-^{(1)}]. \tag{5}$$

whose eigen values are obtained as

$$E_1 = \omega_+, E_2 = \omega_-, E_3 = -\omega_-, E_4 = -\omega_+. \tag{6}$$

with

$$\omega_\pm = \frac{1}{2}(\sqrt{\lambda^2 + F^2} \pm \lambda). \tag{7}$$

As a result, it is obvious that $E_1 > E_2 > E_3 > E_4$.

Furthermore, the eigen states $|E_i\rangle$ which satisfy $H_s|E_i\rangle = E_i|E_i\rangle$ are obtained as

$$(|E_1\rangle, |E_2\rangle, |E_3\rangle, |E_3\rangle))^T = U(|ee\rangle, |eg\rangle, |ge\rangle, |gg\rangle))^T, \tag{8a}$$

$$(|ee\rangle, |eg\rangle, |ge\rangle, |gg\rangle))^T = U^{-1}(|E_1\rangle, |E_2\rangle, |E_3\rangle, |E_3\rangle))^T, \tag{8b}$$

where the unitary transformation $U$ is given by

$$U = \begin{pmatrix} \frac{F}{2G_+} & \frac{\omega_+}{G_+} & \frac{\omega_+}{G_+} & \frac{F}{2G_+} \\ -\frac{F}{2G_-} & \frac{\omega_-}{G_-} & -\frac{\omega_-}{G_-} & \frac{F}{2G_-} \\ \frac{F}{2G_-} & -\frac{\omega_-}{G_-} & -\frac{\omega_-}{G_-} & \frac{F}{2G_-} \\ -\frac{F}{2G_+} & -\frac{\omega_+}{G_+} & \frac{\omega_+}{G_+} & \frac{F}{2G_+} \end{pmatrix}, \tag{9}$$

and the inverse transformation is

$$U^{-1} = U^\dagger = \begin{pmatrix} \frac{F}{2K_+} & -\frac{F}{2K_-} & \frac{F}{2K_-} & -\frac{F}{2K_+} \\ \frac{K_+}{2M} & \frac{K_-}{2M} & -\frac{K_-}{2M} & \frac{K_+}{2M} \\ \frac{K_+}{2M} & -\frac{K_-}{2M} & -\frac{K_-}{2M} & \frac{K_+}{2M} \\ \frac{F}{2K_+} & \frac{F}{2K_-} & \frac{F}{2K_-} & \frac{F}{2K_+} \end{pmatrix}. \tag{10}$$

In the above equations, we have defined $G_\pm = \sqrt{F^2 + \lambda^2 \pm \lambda\sqrt{F^2 + \lambda^2}}$, $K_\pm = \sqrt{F^2 \pm 2\lambda\omega_\pm}$ and $M = \sqrt{F^2 + \lambda^2}$.

In terms of the eigen states $|E_i\rangle$, we will have

$$\begin{aligned} \sigma_x^{(1)} &= |ge\rangle\langle ee| + |ee\rangle\langle ge| + |gg\rangle\langle eg| + |eg\rangle\langle gg| \\ &= \frac{F}{M}(\tau_{11} + \tau_{22} - \tau_{33} - \tau_{44}) + \frac{\lambda}{M}(\tau_{13} - \tau_{24} + \text{H.c.}), \end{aligned} \tag{11}$$

$$\begin{aligned} \sigma_x^{(2)} &= |ee\rangle\langle eg| + |eg\rangle\langle ee| + |gg\rangle\langle ge| + |ge\rangle\langle gg| \\ &= \frac{F}{M}(\tau_{11} - \tau_{22} - \tau_{33} + \tau_{44}) + \frac{\lambda}{M}(\tau_{13} + \tau_{24} + \text{H.c.}), \end{aligned} \tag{12}$$

where $\tau_{ij} = |E_i\rangle\langle E_j|$. Then, performing the rotating wave approximation, the interaction Hamiltonian becomes

$$V \approx \sum_k \frac{g_k\lambda}{M}(\tau_{13}b_k + \tau_{31}b_k^\dagger - \tau_{24}b_k - \tau_{42}b_k^\dagger) + \sum_k \frac{f_k\lambda}{M}(\tau_{13}c_k + \tau_{31}c_k^\dagger - \tau_{24}c_k - \tau_{42}c_k^\dagger). \tag{13}$$

Thanks to the general form of the Markovian master equation

$$\frac{d}{dt}\rho = -\int_0^\infty d\tau \text{Tr}_B[V_I(t), [V_I(t-\tau), \rho \otimes \rho_B]] \tag{14}$$

with $V(t) = \exp[i(H_s + H_B)t]V\exp[-i(H_s + H_B)t]$, we finally reach the Redfield master equation in the Schödinger picture as

$$\frac{d\rho}{dt} = \mathcal{L}\rho = -i[\sum_{i=1}^4 E_i|E_i\rangle\langle E_i|, \rho] + D[\rho], \tag{15}$$

where the dissipator reads

$$D[\rho] = [\Gamma_1(M) + \Gamma_2(M)][2(\tau_{31}\rho\tau_{13} + \tau_{42}\rho\tau_{24}) - (\tau_{11} + \tau_{22})\rho - \rho(\tau_{11} + \tau_{22})]$$
$$[\gamma_1(M) + \gamma_2(M)][2(\tau_{13}\rho\tau_{31} + \tau_{24}\rho\tau_{42}) - (\tau_{33} + \tau_{44})\rho - \rho(\tau_{33} + \tau_{44})]$$
$$-2[\gamma_1(M) - \gamma_2(M)](\tau_{13}\rho\tau_{42} + \tau_{24}\rho\tau_{31}) - 2[\Gamma_1(M) - \Gamma_2(M)][\tau_{31}\rho\tau_{24} + \tau_{42}\rho\tau_{13}].$$

$$(16)$$

with

$$\gamma_i(\omega) = \frac{\lambda^2}{M^2}J_i(M)N_i(M), \ \Gamma_i(\omega) = \frac{\lambda^2}{M^2}J_i(M)\mathcal{N}_i(M). \tag{17}$$

We emphasize that the last line in Eq. (16) comes from the non-secular terms. In the equilibrium situation, that is, $T_1 = T_2$ for boson reservoirs or $T_1 = T_2, \mu_1 = \mu_2$ for fermion reservoirs, these terms will disappear. The same situation has also been found in the two-qubit system, which are immersed in the nonequilibrium reservoirs [52].

## 2.2 Non-resonant driving

In the non-resonant driving case, that is, $\Delta_1 \neq 0$ and/or $\Delta_2 \neq 0$, an analytical solution of the Hamiltonian in Eq. (1) is cumbersome. However, both the eigen energies and the corresponding eigen states, and hence the unitary transformation $U$ can be obtained numerically.

In the eigen states presentation, we have

$$\sigma_x^{(1)} = \sum_{i,j=1}^{4} \chi_{ij}^{(1)}\tau_{ij}, \sigma_x^{(2)} = \sum_{i,j=1}^{4} \chi_{ij}^{(2)}\tau_{ij}. \tag{18}$$

where $\chi_{ij}^{(m)} = \langle E_i|U\sigma_x^{(m)}U^\dagger|E_j\rangle$ for $m = 1, 2$. Under the rotating wave approximation, the interaction Hamiltonian between the charger-battery system and the reservoirs can be written as

$$V = \sum_{i,j>i}\sum_k \left(g_k\chi_{ij}^{(1)}b_k\tau_{ij} + f_k\chi_{ij}^{(2)}c_k\tau_{ij} + \text{H.c.}\right), \tag{19}$$

where we have ordered the eigen energies by $E_1 > E_2 > E_3 > E_4$. Consequently, the master equation under Markovian approximation but beyond secular approximation reads

$$\frac{d}{dt}\rho = \mathcal{L}\rho = -i[\sum_{j=1}^{4}E_j|E_j\rangle\langle E_j|, \rho] + \mathcal{D}_1(\rho) + \mathcal{D}_2(\rho), \tag{20}$$

where

$$\mathcal{D}_1(\rho) = \sum_{i,j>i}\sum_{m,n>m}\sum_{\alpha=1,2} J_\alpha(\epsilon_{mn})\mathcal{N}_\alpha(\epsilon_{mn})\mathcal{G}_{ij,mn}^{(\alpha)}[\rho], \tag{21}$$

$$\mathcal{D}_2(\rho) = \sum_{i,j>i}\sum_{m,n>m}\sum_{\alpha=1,2} J_\alpha(\epsilon_{mn})N_\alpha(\epsilon_{mn})\mathcal{H}_{ij,mn}^{(\alpha)}[\rho], \tag{22}$$

and

$$\mathcal{G}_{ij,mn}^{(\alpha)}[\rho] = \chi_{ji}^{(\alpha)}\chi_{mn}^{(\alpha)}\left(\tau_{ji}\rho\tau_{mn} - \rho\tau_{mn}\tau_{ji}\right) + \chi_{ij}^{(\alpha)}\chi_{nm}^{(\alpha)}\left(\tau_{nm}\rho\tau_{ij} - \tau_{ij}\tau_{nm}\rho\right), \tag{23}$$

$$\mathcal{H}_{ij,mn}^{(\alpha)}[\rho] = \chi_{ij}^{(\alpha)}\chi_{nm}^{(\alpha)}\left(\tau_{ij}\rho\tau_{nm} - \rho\tau_{nm}\tau_{ij}\right) + \chi_{ji}^{(\alpha)}\chi_{mn}^{(\alpha)}\left(\tau_{mn}\rho\tau_{ji} - \tau_{ji}\tau_{mn}\rho\right). \tag{24}$$

In the above equations, we have defined $\epsilon_{ij} = E_i - E_j$ as the energy level spacing between the states $|E_i\rangle$ and $|E_j\rangle$.

In the eigen states representation, the initial state is obtained as $|\psi(0)\rangle_e = U^\dagger |\psi(0)\rangle$, and the system undergoes the time evolution which is governed by the master equation in Eq. (16) or Eq. (20), depending on whether the driving to the charger is in resonance. After a time interval $\tau$, at which the charging process is supposed to be ended, the system will reach a state with density matrix $\rho(\tau)$. Back to the bare representation and Schödinger picture, we will have

$$\tilde{\rho}(\tau) = U_1(\tau) U \rho(\tau) U^\dagger U_1^\dagger(\tau), \tag{25}$$

with $U_1(\tau) = \exp[i(\sigma_z^{(1)} + \sigma_z^{(2)})\tau]$.

We formally assume that $\tilde{\rho}(\tau)$ can be expressed as (in the basis of $\{|ee\rangle, |eg\rangle, |ge\rangle, |gg\rangle\}$)

$$\tilde{\rho}(\tau) = \begin{pmatrix} M_{11} & M_{12} & M_{13} & M_{14} \\ M_{21} & M_{22} & M_{23} & M_{24} \\ M_{31} & M_{32} & M_{33} & M_{34} \\ M_{41} & M_{42} & M_{43} & M_{44} \end{pmatrix}, \tag{26}$$

the reduced density matrix for the battery subsystem can be expressed as (in the basis of $\{|e\rangle, |g\rangle\}$)

$$\rho_B = \begin{pmatrix} M_{33} + M_{11} & M_{12} + M_{34} \\ M_{21} + M_{43} & M_{44} + M_{22} \end{pmatrix}. \tag{27}$$

In the quantum battery scenario, one of the quantities we are interested is the mean charging energy $E_B$ contained in the battery at the end of the charging process, which is expressed as [59, 60]

$$E_B(\tau) = \text{Tr}[H_B \rho_B], \tag{28}$$

with $H_B = \omega |e\rangle\langle e|$. Another quantity is the ergotropy function $\mathcal{E}_B$, whose physical meaning can be understood as follows.

Considering the state of a quantum system, which is characterized by the free Hamiltonian $H$, is given by the density matrx $\rho$. In terms of spectrum decomposition, $\rho$ and $H$ can be written as

$$\rho = \sum_n r_n |r_n\rangle\langle r_n|, H = \sum_n e_n |e_n\rangle\langle e_n|, \tag{29}$$

where $r_n$ and $e_n$ are the eigen values of $\rho$ and $H$ with corresponding eigen states $|r_n\rangle$ and $|e_n\rangle$. Grouping the eigen energies as $r_0 \geq r_1 \geq r_2 \geq \cdots$ and $e_0 \leq e_1 \leq e_2 \leq \cdots$, we can construct a quantum state, whose density matrix yields

$$\rho^{(p)} = \sum_n r_n |e_n\rangle\langle e_n|. \tag{30}$$

When a quantum system is in such a state, it is unable to release energy to the outside, so, the state $\rho^{(p)}$ is called passive state [59, 60]. The energy of the passive state is given by

$$E^{(p)} = \text{Tr}(H \rho^{(p)}) = \sum_n r_n e_n, \tag{31}$$

which corresponds to

$$E^{(p)} = \min_U \text{Tr}[H U \rho_B(\tau) U^\dagger], \tag{32}$$

Here, the minimization is performed on all of the local unitaries $U_B$ on the battery subsystem, and this corresponds to the part energy in $E_B(\tau)$ that is locked into the correlations in the

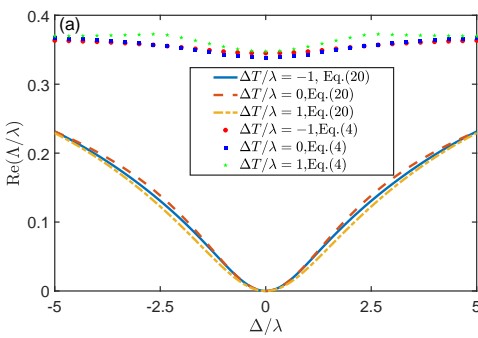 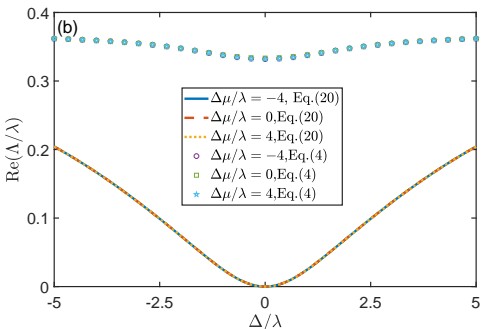

Figure 2: Liouvillian gap of the system as a function of $\Delta = \Delta_1 - \Delta_2$ for (a) boson and (b) fermion reservoirs for the Ohmic spectra. The parameters are set as $F = 0.5\lambda, \omega_c = 5\lambda, \alpha_1 = \alpha_2 = 0.1\lambda, \bar{\Delta} = (\Delta_1 + \Delta_2)/2 = 0$ and (a) $\bar{T} = (T_1 + T_2)/2 = \lambda$ and (b) $T_1 = T_2 = \lambda, \bar{\mu} = (\mu_1 + \mu_2)/2 = 2\lambda$.

system, preventing one from accessing it via local operations on the battery. Therefore, the ergotropy, which is actually the available energy that can be extracted from the battery is

$$\mathcal{E}_B(\tau) = E_B(\tau) - E^{(p)}. \tag{33}$$

For our two-level battery system, the above qualities can be obtained as [28]

$$E_B(\tau) = \frac{\omega}{2}(N+1), \tag{34}$$

$$\mathcal{E}_B(\tau) = \frac{\omega}{2}\left(\sqrt{\langle\sigma_z\rangle^2 + 4\langle\sigma_+\rangle\langle\sigma_-\rangle} + \langle\sigma_z\rangle\right)$$

$$= \frac{\omega}{2}\left(\sqrt{N^2 + 4|\mathcal{N}|^2} + N\right), \tag{35}$$

where $N = M_{11} + M_{33} - M_{22} - M_{44}$ and $\mathcal{N} = M_{12} + M_{34}$.

In what follows, we will discuss the characterization of the Liouvillian and the charging efficiency $P(\tau) = \mathcal{E}_B(\tau)/E_B(\tau)$. We will set $\tau = 20000/\lambda$ when we discuss the quantities on the steady state. We have checked that the system has achieved the steady state after this time interval.

## 3   Bistability and steady state entanglement

In the last section, we have listed the phenomenological master equation and derived the Redfield master equation for the driven charger-battery system. The master equations in Eqs. (15,20) show that the Liouvillian superoperator $\mathcal{L}$ belongs to the Liouvillian space $\mathcal{H} \otimes \mathcal{H}$, where $\mathcal{H}$ is the Hilbert space of the quantum system. To fully determine the dynamics of the system, one has to resort to the spectrum of the superoperator $\mathcal{L}$, with

$$\mathcal{L}\rho_i = \lambda_i \rho_i. \tag{36}$$

Equivalently, $\rho_i$ is an eigen right vector of $\mathcal{L}$ with eigen value $\lambda_i$. It can be proven that $\mathrm{Re}[\lambda_i] \leq 0$ for arbitrary $i$. We sort the eigen values $\lambda_i$ in a way that $\mathrm{Re}[\lambda_0] < \mathrm{Re}[\lambda_1] < \cdots < \mathrm{Re}[\lambda_n]$, so $\mathrm{Re}[\lambda_0] = 0$ corresponds to the steady state $\rho_{ss} = \rho_0/\mathrm{Tr}(\rho_0)$ and the Liouvillian gap $\Lambda = \mathrm{Re}[\lambda_1]$ is also called the asymptotic decay rate [61], which determines the slowest relaxation dynamics in the long-time limit.

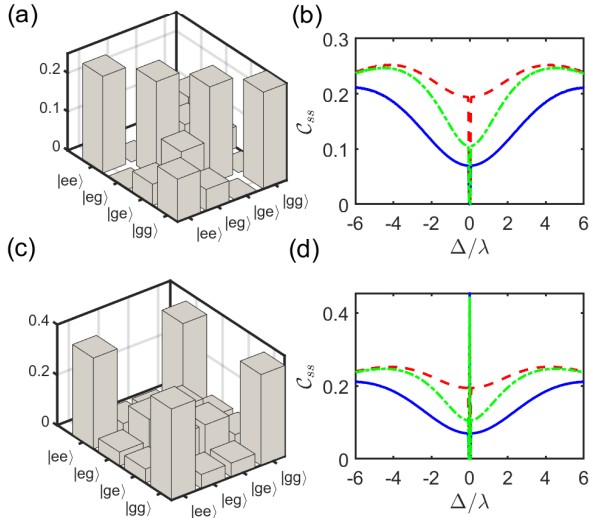

Figure 3: Tomography [(a) and (c)] and concurrence [(b) and (d)] of the steady states governed by our Redfield master equation under the nonequilibrium boson reservoirs. The initial state is set as $|\psi(0)\rangle = |eg\rangle$ for (a) and (b) and $|\psi(0)\rangle = (|eg\rangle + |ge\rangle)/\sqrt{2}$ for (c) and (d). The parameters are set as $F = 0.5\lambda, \omega_c = 5\lambda, \bar{T} = \lambda, \alpha_1 = \alpha_2 = 0.1\lambda$. In (a) and (c), we set $\Delta T = \lambda$. In (b) and (d), we set $\Delta T = 0$ for blue solid curve, $\Delta T/\lambda = 1$ for red dashed curve and $\Delta T/\lambda = -1$ for green dotted-dashed curve.

The results of Liouvillian gap are demonstrated in Fig. 2 (a) and (b) for the boson and fermion reservoirs by choosing the Ohmic spectrum with

$$J_i(\omega) = \alpha_i \omega \exp(-\omega/\omega_c), \tag{37}$$

for $i = 1, 2$. The results show that the Liouvillian gap behaves similarly for different non-equilibrium natures, that is, the temperature difference $\Delta T = T_1 - T_2$ for the boson reservoirs and the chemical potential difference $\Delta \mu = \mu_1 - \mu_2$ for the fermion reservoirs. The results given by the phenomenological master equation [Eq. (4)] show that the Liouvillian gap is always nonzero, which implies that there is only one steady state as the evolution time tends to be infinity. On the contrary, for the Redfield master equation [Eq. (20)], it is found that the Liouvillian gap closes when $\Delta_1 = \Delta_2 = 0$, that is, the charger and the battery are identical, and the charger is driven in resonance. In this case, the system exhibits bistability. It is then constructive to investigate the steady state when the charger-battery system is prepared in different initial states.

Let us firstly consider that the system is prepared in the initial state $|\psi(0)\rangle = |eg\rangle$, that is, the charger is in the excited state while the battery is in the ground state. A tomography for the density matrix $|\rho_{ij}|$ for $\Delta_1 = \Delta_2 = 0$ is shown in Fig. 3(a) for the boson reservoirs case. It shows that, the population exhibits an uniform distribution and there exists some minor coherence. For the same initial state, we plot the steady state concurrence $\mathcal{C}$ [62,63] in Fig. 3 (b), which quantifies the entanglement for a two two-dimension composite quantum system as a function of $\Delta$ for $\bar{\Delta} = 0$. We observe $\mathcal{C} = 0$ at $\Delta = 0$, in which the charger is driven in resonance. Away from the resonant driving $\Delta \neq 0$, a non-zero entanglement emerges and the non-equilibrium nature of the reservoir can enhance the steady entanglement. As shown in Fig. 3 (b), the values of $\mathcal{C}$ for $\Delta T/\lambda = 1$ (red dashed curve) and $\Delta T/\lambda = -1$ (green dotted-dashed curve) are higher than that for $\Delta T = 0$ (blue solid curve) for the non-resonant driving case.

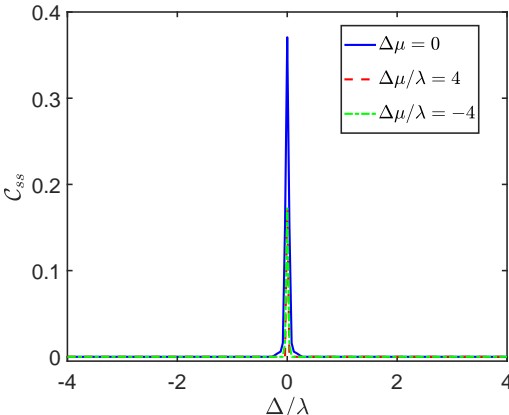

Figure 4: Concurrence of the steady states governed by our Redfield master equation under the fermion reservoirs. The initial state is set as $|\psi(0)\rangle = (|eg\rangle + |ge\rangle)/\sqrt{2}$. The parameters are set as $F = 0.5\lambda, \omega_c = 5\lambda, T_1 = T_2 = \lambda, \bar{\mu} = 2\lambda, \alpha_1 = \alpha_2 = 0.1\lambda$.

However, the entanglement will be recovered in the resonant driving by preparing the initial state as $|\psi(0)\rangle = (|eg\rangle + |ge\rangle)/\sqrt{2}$. In this case, the tomography and concurrence are given in Fig. 3(c) and (d), respectively. Under the resonant driving, we observe $|\rho_{ee,ee}| = |\rho_{gg,gg}| = |\rho_{ee,gg}|$, $|\rho_{eg,eg}| = |\rho_{ge,ge}| = |\rho_{eg,ge}|$ and $|\rho_{ee,eg}| = |\rho_{ee,ge}| = |\rho_{gg,eg}| = |\rho_{gg,ge}|$, and this symmetry leads to a non-zero steady state entanglement for $\Delta = 0$ as shown in Fig. 3(d). Similar to Fig. 3(b), the non-equilibrium also increases the entanglement in the regime of $\Delta \neq 0$.

For the resonant driving, the steady states for the equilibrium and non-equilibrium fermion reservoirs are similar to that for the boson reservoirs, and we will not give the tomography results here. However, in the non-resonant driving situation, the steady state entanglement is dramatically different for the fermion and boson reservoirs. When the charger-battery system is prepared in the separated state $|eg\rangle$, the concurrence for the steady state is smaller than 0.01 even under the resonant driving. For the initial state $|\psi(0)\rangle = (|eg\rangle + |ge\rangle)/\sqrt{2}$, although it still yields a zero concurrence under the non-resonant driving, the resonant driving can induce a considerable entanglement as shown in Fig. 4 for the fermion reservoirs. Furthermore, the non-equilibrium induced by the chemical potential difference between the two fermion reservoirs weakens the entanglement, it is opposite of the case for the boson reservoirs, in which the non-equilibrium is supplied by the temperature difference.

## 4   Efficiency of quantum battery for resonant driving

In this section, we will discuss the efficiency of the quantum battery based on the Redfield master equation in Eq. (16) in the resonant driving situation. We will also give a comparison to the results based on the phenomenological master equation in Eq. (4).

We first prepare the system in the state $|\psi(0)\rangle = \cos\theta|eg\rangle + \sin\theta e^{i\phi}|ge\rangle$ to discuss the efficiency of quantum battery model in the equilibrium boson reservoirs. Since the phenomenological master equation in Eq. (4) yields a single steady state, we can obtain an efficiency of $P_{ss} \approx 0.94$ for arbitrary $\theta$ and $\phi$. However, based on the Redfield master equation in Eq. (16), we contour plot the dependence of $P_{ss}$ on $\theta$ and $\phi$ in Fig. 5. It shows that, for the separated states ($\theta = 0, \pi/2$ and $\pi$), the efficiency is always zero. For the maximum entanglement initial state $\theta = \pi/4$ or $\theta = 3\pi/4$, the efficiency is sensitive to the phase $\phi$. For $\phi = \pi/2$, which yields $|\psi(0)\rangle = (|eg\rangle \pm i|ge\rangle)/\sqrt{2}$, the efficiency is still zero. However, for $\phi = 0$ or $\phi = \pi$, that is $|\psi(0)\rangle = (|eg\rangle \pm |ge\rangle)/\sqrt{2}$, we can achieve a maximum efficiency by $P_{ss} \approx 0.23$. Actually, as

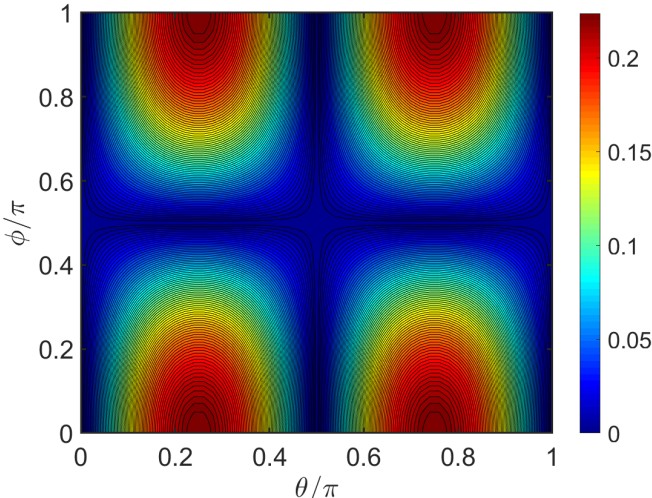

Figure 5: Efficiency $P_{ss}$ as function of $\theta$ and $\phi$ for the initial state $|\psi(0)\rangle = \cos\theta|eg\rangle + \sin\theta e^{i\phi}|ge\rangle$ in the bosonic reservoir. The parameters are set as $F = 0.5\lambda, \omega_c = 5\lambda, \Delta_1 = \Delta_2 = 0, T_1 = T_2 = \lambda, \alpha_1 = \alpha_2 = 0.1\lambda$.

shown before, under these initial states, the system will evolve to an entangled state, which implies that the entanglement may enhance the efficiency of the quantum battery.

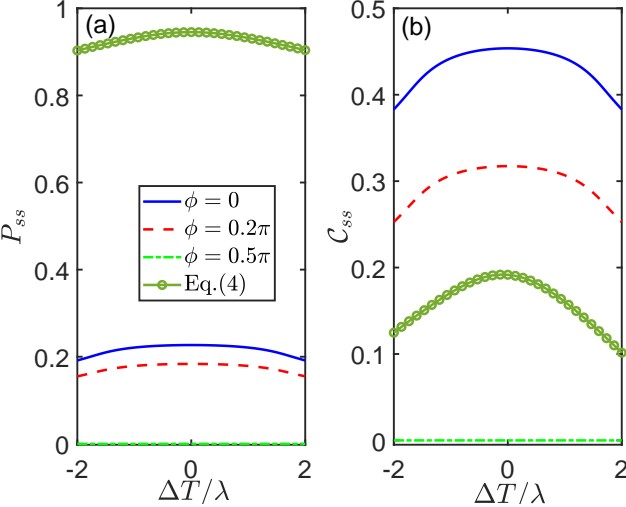

Figure 6: (a) Efficiency $P_{ss}$ and (b) concurrence $\mathcal{C}_{ss}$ as functions of $\phi$ for the initial state $|\psi(0)\rangle = (|eg\rangle + e^{i\phi}|ge\rangle)/\sqrt{2}$ in the nonequilibrium boson reservoir. The parameters are set as $F = 0.5\lambda, \omega_c = 5\lambda, \Delta_1 = \Delta_2 = 0, \bar{T} = \lambda, \alpha_1 = \alpha_2 = 0.1\lambda$.

This entanglement enhanced efficiency is also verified in the non-equilibrium case. As shown in Fig. 6, where the steady state efficiency and the concurrence are plotted as functions of $\Delta T$, the behaviors of the efficiency and the entanglement are positively related to each other in our system. For example, both of the entanglement and the efficiency reach their maximum values at the equilibrium situation with $\Delta T = 0$ for $\theta \neq 0.5\pi$, which otherwise keeps zero. On the other hand, the results based on the phenomenological master equation in Eq. (4) show its independence on $\phi$ due to its single steady state. Although the efficiency based on Eq. (4) is higher than that based on Eq. (15), the steady state entanglement is not as shown in Fig. 6

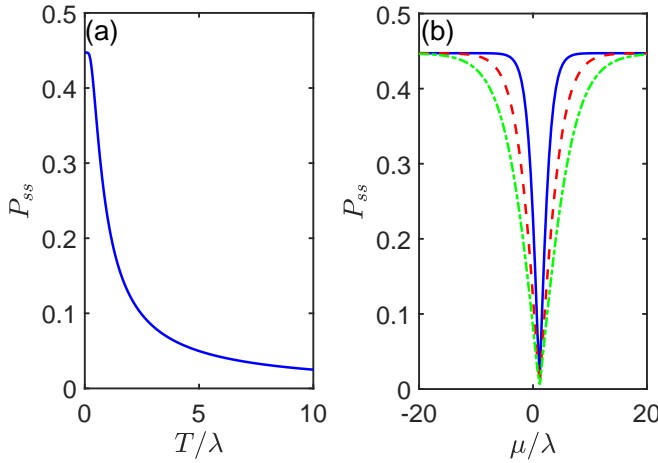

Figure 7: Efficiency $P_{ss}$ as function of $T_1 = T_2 = T$ in boson reservoirs (a) and $\mu_1 = \mu_2 = \mu$ in fermion reservoirs (b) with the initial state being $|\psi(0)\rangle = (|eg\rangle + |ge\rangle)/\sqrt{2}$. The parameters are set as $F = 0.5\lambda, \omega_c = 5\lambda, \Delta_1 = \Delta_2 = 0, \alpha_1 = \alpha_2 = 0.1\lambda$. In (b), we set $T_1 = T_2 = T = \lambda$ for the blue solid curve, $T = 2\lambda$ for red dashed curve and $T = 3\lambda$ for green dotted-dashed curves respectively.

(b), this is because we have resorted to different treatment.

The above results for the boson reservoirs are also valid for the fermion reservoirs. For example, the initial state $|\psi(0)\rangle = (|eg\rangle \pm |ge\rangle)/\sqrt{2}$ leads to the highest efficiency in our quantum battery setup and this efficiency will reach its maximum value in the equilibrium situation, that is, $T_1 = T_2$ for the boson reservoirs and $T_1 = T_2, \mu_2 = \mu_2$ for the fermion reservoirs. In Fig. 7(a), we plot the efficiency as a function of the temperature $T_1 = T_2 = T$ in the equilibrium boson reservoir by setting the initial state as $|\psi(0)\rangle = (|eg\rangle + |ge\rangle)/\sqrt{2}$. It is not surprising that the high temperature will suppress the efficiency due to the weakening of the quantum nature by the thermal effect. The battery efficiency for fermion reservoirs as a function of $\mu_1 = \mu_2 = \mu$ with different $T$ is plotted in Fig. 7(b), which shows that the efficiency drops off for a fixed $\mu$, which is not dependent of $T$. Moreover, as $\mu$ is positively or negatively higher enough, the efficiency will be saturated at a relatively large value for arbitrary temperature. It means that coupling to fermion reservoirs exchanging particles with the system is more robust to the thermal effect than boson reservoirs exchanging energy with the system, to enhance the quantum battery performance.

## 5 Efficiency of quantum battery for non-resonant driving

In this section, we study the efficiency of the quantum battery when the external driving to the charger is not in resonance and the charger and the battery are not identical, that is, $\Delta_1 \neq \Delta_2$. In this case, the Liouvillian gap of the Redfield master equation opens, and there is only one steady state. In the following discussions, we prepare the initial state as $|\psi(0)\rangle = |eg\rangle$, and investigate the battery efficiency in both the equilibrium and nonequilibrium cases.

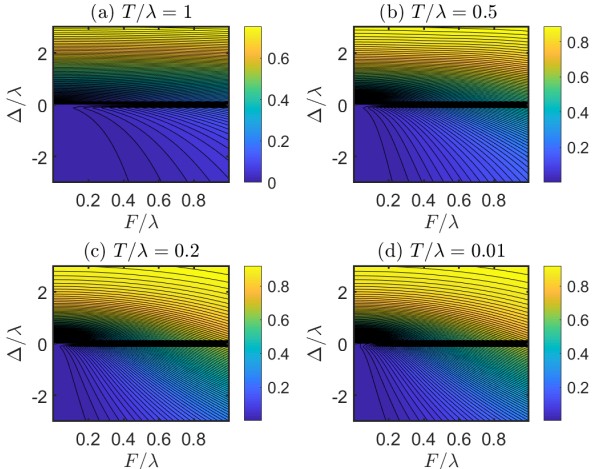

Figure 8: The efficiency $P_{ss}$ as a function of the driving strength $F$ and the detuning $\Delta$ for the boson equilibrium reservoir. The parameters are set as $\omega_c = 5\lambda, \bar{\Delta} = 0, \alpha_1 = \alpha_2 = 0.1\lambda$.

## 5.1 Boson reservoirs

In Fig. 8, we plot the efficiency $P_{ss}$ as a function of the driving strength $F$ and the detuning $\Delta$ for different temperatures $T_1 = T_2 = T$ for boson reservoirs. One sees that the efficiency can reach its maximum value for $\Delta > 0$. In this case, the nonequilibrium is supplied by the driving to the charger and the dissipation for both of the charger and the battery. As a result, the efficiency is enhanced by a compensation mechanism, in which the frequency of the driven charger should be higher than that of the battery. The results also show that a lower temperature is helpful to broaden the high efficiency regime.

For the non-equilibrium boson reservoirs, we demonstrate the dependence of the battery efficiency $P_{ss}$ on the charger-battery detuning $\Delta = \Delta_1 - \Delta_2$ and the temperature difference $\Delta T = T_1 - T_2$ in Fig. 9. One sees that the efficiency has higher values in the top right corner, in which the efficiency is 0.93, being much higher than that with resonant driving ($\Delta = 0$). In the top right corner where $\Delta > 0$ and $\Delta T > 0$, that is, $\omega_1 > \omega_2$ and $T_1 > T_2$. This suggests that, to enhance the performance of the quantum battery, it is beneficial to have the frequency of the battery to be lower than that of the charger and couple it to the reservoir with lower temperature. Meanwhile, the regime with high efficiency shrinks with the increase of $\bar{T}$, which implies that a higher temperature is harmful for the performance of the quantum battery.

## 5.2 Fermion reservoirs

In Figs. 10 and 11, we plot the efficiency $P_{ss}$ for the fermion reservoirs with same and different chemical potentials respectively. For the same chemical potential $\mu_1 = \mu_2 = \mu$, the results in Fig. 10 show that the maximum efficiency can be obtained by setting $\Delta < 0$ for $\mu > 0$ and $\Delta > 0$ for $\mu < 0$. For the nonequilibrium fermion reservoirs, the efficiency shows a complicated dependence on the chemical potential difference and detuning for different average chemical potential $\bar{\mu} = (\mu_1 + \mu_2)/2$. For small $\bar{\mu}$, for example $\bar{\mu} = 0$ and $\bar{\mu} = 3\lambda$, one observes that the maximum efficiency can be reached in the parameter regime for either $\Delta > 0$ or $\Delta < 0$ in Fig. 11(a) and (b). For a positively large $\bar{\mu} = 6\lambda$, as shown in Fig. 11(c), the optimal regime mainly locates at $\Delta < 0$. As a dramatic contrast, this optimal regime is transferred to that with $\Delta > 0$ when $\bar{\mu} = -6\lambda$. These behaviors can be physical intuitively explained by the compensation mechanism. When the chemical potential is negative, the charger-battery system is easy

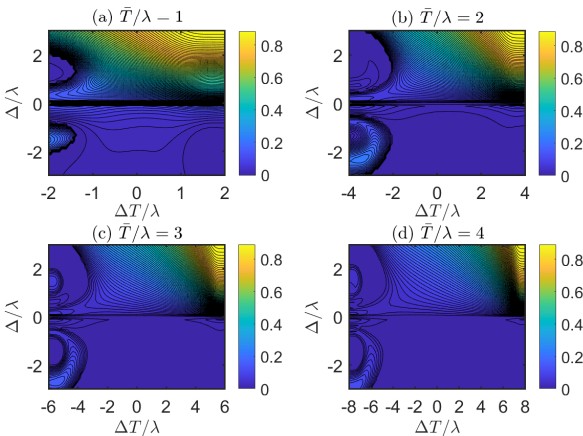

Figure 9: The efficiency $P_{ss}$ as a function of $\Delta T$ and $\Delta$ for nonequilibrium boson reservoirs. The parameters are set as $F = 0.5\lambda, \omega_c = 5\lambda, \bar{\Delta} = 0, \alpha_1 = \alpha_2 = 0.1\lambda$.

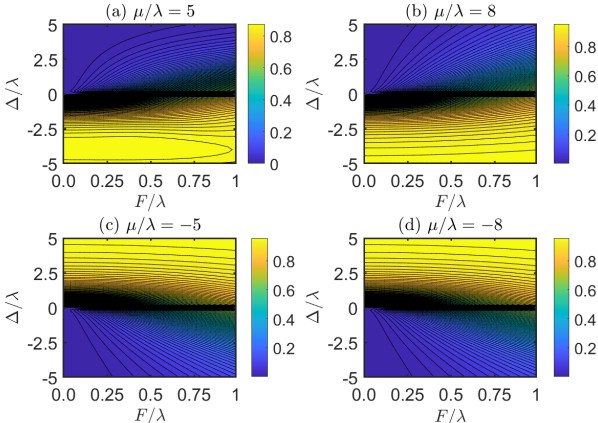

Figure 10: The efficiency $P_{ss}$ as a function of the driving strength $F$ and the detuning $\Delta$ for the fermion equilibrium reservoirs. The parameters are set as $T_1 = T_2 = \lambda, \omega_c = 5\lambda, \bar{\Delta} = 0, \alpha_1 = \alpha_2 = 0.1\lambda$.

to release particles to the environments. In this case, the charger should be higher than the battery in frequency, to compensate the particle loss to achieve a larger efficiency. Vice verse, when the chemical potential is positive, the frequency of the charger should be lower than the battery. Therefor, we reach the highest efficiency in the regime $\Delta > 0$ for $\bar{\mu} < 0$ as shown in Figs. 10(c),(d) and Fig. 11(d). On the contrary, the high efficiency regime can be found with $\Delta < 0$ in Figs. 10(a),(b) and Fig. 11(c) when $\bar{\mu} > 0$.

Furthermore, for small $\bar{\mu}$ and $\Delta\mu$, the particle flow for the charger and the battery can be either the same (both of them release or absorb particles) or different (one partner of charger-battery system releases particles while the other absorbs). Therefore, the highest efficiency can appear in both the $\Delta > 0$ regime and the $\Delta < 0$ regime as shown in Figs. 11 (a) and (b).

The above results for both of the boson and fermion reservoirs demonstrate the advantages of non-resonant driving setup in the following two aspects. First, the value of the highest efficiency in Figs. 8 to 11 for non-resonant driving is about $3 - 4$ times higher than that in Fig. 6 for resonant driving. Second, since the Redfield master equation yields a unique steady state under the non-resonant driving, there is no need to prepare the charger-battery system in the initial entangled state to achieve a high efficiency. This simplifies the experimental

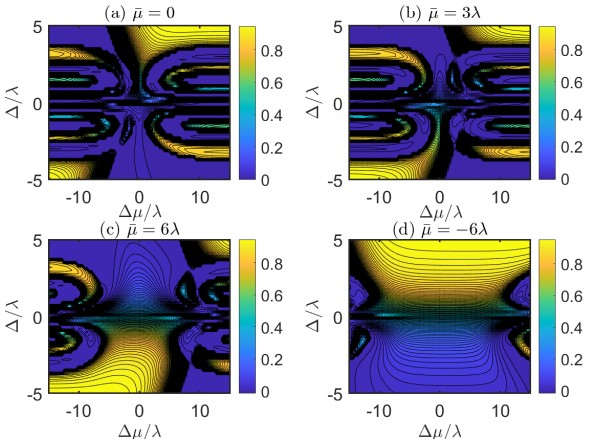

Figure 11: The efficiency $P_{ss}$ as a function of $\Delta\mu$ and $\Delta$ for nonequilibrium fermion reservoirs. The parameters are set as $F = 0.5\lambda, T_1 = T_2 = \lambda, \omega_c = 5\lambda, \bar{\Delta} = 0, \alpha_1 = \alpha_2 = 0.1\lambda$.

difficulties in entangled state preparation.

# 6 Conclusion

In this paper, we have investigated the efficiency of a quantum battery setup, where both the charger and the battery are two-level systems. They coherently couple to each other and simultaneously interact with their individual boson or fermion reservoirs, with nonequilibrium characterized by the temperature or chemical potential difference. By taking the external driving to the charger beyond the traditional phenomenological manner, we obtain the Redfield master equation without secular approximation. When the transition frequencies of the charger and the battery are equal to each other and the charger is driven in resonance, we show that the system will exhibit a bistability behavior which is determined by the closed Liouvillian gap and can not be predicted by the conventional Lindblad master equation. As a result, the on demand chosen initial entangled state may lead to a relatively higher efficiency for the quantum battery. This efficiency can be furthermore enhanced in the non-resonant driving cases. For example, for both the boson reservoirs and fermion reservoirs, one can found a maximum efficiency which achieves 93%, being much higher than that for resonant driving.

We also show the role of the nonequilibrium of the reservoirs in the performance of the quantum battery setup. When the charger is driven in resonance, the nonequilibrium reduced the efficiency of the quantum battery for both the boson and fermion reservoirs. However, when the charger is driving nonresonantly, a temperature difference for boson reservoirs and chemical potential difference for fermion reservoirs can enhance the quantum battery efficiency in a compensation manner. In this sense, the nonequilibrium reservoirs provides us an effective approach for designing energy devices based on open system.

**Author contributions** Z. Wang and J. Wang designed the research. Z. Wang and H. Yu performed the main calculations. Z. Wang and J. Wang wrote the manuscript, All of the authors contributed to the planning and made revision of the manuscript.

**Funding information**    National Key R&D Program of China (Grant No. 2021YFE0193500). Science and Technology Development Project of Jilin Province (Grant No. 20230101357JC) and National Natural Science Foundation of China (Grants No. 21721003 and 12234019).

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
