# Peer review of "Driven-dissipative quantum battery with nonequilibrium reservoirs"

_SciPost Physics_

## Round 1 · Referee Report · Anonymous · 2023-12-4

Report

In this work the authors study several improvements over the standard scenario of a qubit quantum battery charged by a driven-dissipative qubit ancilla. In particular, they add a different noise on the battery and they treat the whole model using a microscopically sound approach. They derive the corresponding Redfield equation at weak coupling and numerically solve it to study the behavior of the efficiency in various parameter regimes. I really liked how they sticked to a more realistic steady-state charging protocol (instead of the classical switch on/off), and their promising results about how nonequilibrium features can enhance the efficiency in a less traditional nonresonant driving scheme. However, before expressing my final recommendation I would like the authors to address the following remarks.

General remarks

1. It is not entirely clear from the presentation how the rotating frame transformation at frequency $\omega_d$ interacts with the rotating wave approximation. Specifically, from Eqs. (1)-(2) I can guess that the starting universe Hamiltonian has been rotated with something like $\exp[i\omega_d( \sigma_z^{(1)} + \sigma_z^{(2)} )t/2 + i\omega_d \sum_k ( b_k^\dagger b_k + c_k^\dagger c_k )t]$, but to me this is not a completely trivial fact and should be specified. This is perhaps partially mentioned below Eq. (25), but a factor $\omega_d/2$ is missing there, and the reservoir part obviously do not appear. If this is the correct transformation, then I believe the expression Eq. (3) for the system-reservoir coupling $V$ is not the rotated one, since counter-rotating terms should present a factor $\exp(2i\omega_d t)$. This does not impact the results of the paper, since the rotating wave approximation is performed from Sec. 2.1 onward: as a consequence, I suggest to remove counter-rotating terms from Eq. (3) and stick with the rotating-wave form throughout, unless I missed some use of counter-rotating terms in the rest of the paper.

2. I do not understand why the constants $K_\pm$ are introduced below Eq. (10), since they are identical to the $G_\pm$ constants. Since $U^{-1} = U^\dagger$, why not use the same notations of Eq. (9)? Moreover, it should be true that $\omega_\pm/G_\pm = K_\pm/2M$, but it is not immediately clear to me that such a relation holds.

3. Eq. (14) shows the general form of the Redfield equation in interaction picture, while Eq. (15) shows its rielaborated form in Schroedinger picture, but the same symbol $\rho$ for the system state is used: I suggest to differentiate the notation to avoid confusion. Moreover, Eq. (14) is written with an interaction $V$ that is $\omega_d$-rotated, but it is not clear if Eq. (15) is rotated as well or not. I guess it is still in the rotated frame, but since in Sec. 2.2 a transformation that inverts the $\omega_d$ rotation is introduced, some clarifications can avoid misunderstanding. I would also add a citation to Breuer-Petruccione just above Eq. (14) and explicitly write that the Redfield equation is obtained under a weak-coupling assumption.

4. A similar notation problem happens in Eq. (28), where the symbol $H_B$ appears to refer to the battery Hamiltonian, even though the same symbol is used in Eq. (2) for the reservoir Hamiltonian. Moreover, it is stated that $H_B = \omega |e\rangle \langle e|$, but with the previous notations it should be $\omega_2/2 (|e \rangle \langle e| - |g \rangle \langle g|)$. I believe this energy shift does not influence the results, but uniformization can make the paper more clear.

5. Below Eq. (37) it is stated that the concurrence is used to measure the stead-state entanglement between charger and battery. I would briefly mention the expression you used to calculate it, just to make the paper more self-contained.

6. The Conclusion section beautifully summarizes the obtained results, but my feeling is that the same cannot be said about the abstract. The one submitted to arXiv is written considerably better. In both versions, however, it is not clear that the "compensation mechanism" the authors mention is related to having temperature or chemical potential difference: in my view, the efficiency enhancement that nonequilibrium features bring to the nonresonant driving scheme is the most important result of the paper and should be emphasized more in the abstract.

Follow-up questions

7. It is interesting to see how assuming resonant driving opens up a nontrivial steady-state space. In Lindblad theory, the closure of the Liouvillian gap is often associated with dissipative phase transitions (see Ref. [58]). I guess some subtleties should be taken into consideration when looking at the Redfield equation, but do you think it is possible to interpret the bistability you obtain using arguments from DPTs and symmetry breaking? A related question is why the steady-state concurrence in Fig. 3 appears to be so discontinuous as a function of $\Delta$.

8. In Fig. 7b you show how the efficiency at equilibrium with resonant driving drops for a certain value of $\mu$ that is not dependent on $T$. Do you have an interpretation for why this happens and/or why it happens at that specific value of $\mu$?

9. It is a known fact in the theory of open quantum systems that the Redfield equation is not positive, meaning that it could lead to the prediction of meaningless negative probabilities for measurement outcomes. Did you have problems in this sense, or your values for $\alpha_{1,2}$ were sufficiently low to not encounter issues?

Writing issues

10. There is a nonnegligible number of grammatical and orthographic issues. For example: "nonequilibrium" is misspelled several times; "eigen states" and "eigen values" instead of "eigenstates" and "eigenvalues"; other statements are ill-formed, especially in the Introduction (e.g., "the quantum battery which is subject to the open system"). There is also a piece of the SciPost template the beginning of the Introduction. I suggest to perform a complete writing recheck.

11. Related to the previous point, I noticed typos in some mathematical expressions. Below Eq. (4), it should be $\delta(\omega - \omega_{bk})$ instead of $\delta(\omega - \omega_1)$ and $\delta(\omega - \omega_{ck})$ instead of $\delta(\omega - \omega_2)$. In Eq. (8), $E_3$ is repeated twice, whereas $E_4$ should appear instead in second occurrences. Below Eq. (25), $\omega_d/2$ should appear in $U_1(\tau)$ I believe.

12. Few suggestions about the figures: in Fig. 2 the label $\Re[\Lambda/\lambda]$ is used to indicate the Liouvillian gap but in the text the notation $\Lambda = \Re[\lambda_1]$ is used; in Fig. 3 maybe it should be specified in the caption that the tomography is performed at $\Delta = 0$; in Fig. 4 the majority of the information is concentrated around $\Delta = 0$ but the plot is too zoomed out to appreciate the chemical potential differences; Figs. 9-11 are maybe too small.

---

## Round 1 · Referee Report · Anonymous · 2024-1-3

Strengths

1. The authors extend their analysis beyond the scope of a Lindblad master equation by employing the Redfield master equation.

Weaknesses

1. The manuscript's presentation quality and use of English are inadequate.
2. The analysis within the manuscript lacks depth, necessitating further investigation.
3. The manuscript does not convincingly explain why the described setup qualifies as a "battery."

Report

This manuscript presents an investigation into an open quantum battery system, consisting of a qubit coupled to a driven qubit (referred to as the charger) and in contact with two thermal reservoirs, both fermionic and bosonic. The study utilizes both Lindblad and Redfield master equations, exploring phenomena such as bistability.

In my opinion, this manuscript fails to do an appropriate analysis due to several important shortcomings.  I find this manuscript to be poorly written and I have found important typos. I believe this article could be published in SciPost Physics only after a serious and major revision.

Here are the main points that need addressing:
1. I am not persuaded that the authors are studying something that could be called a "battery". The authors study the steady state of an out-of-equilibrium problem and report the charging efficiency. It is not clear, however, how this system, is continuously driven by the baths and the classical field, could operate as a battery, storing energy for a long time and providing it to a third system to do something useful.

2. The manuscript defines efficiency as the ratio between ergotropy and average energy. think it would be much better to show also the average energy, as the efficiency alone is not indicative of the energy injected in the battery. As an example, if the battery has a very low amount of energy, but the battery's state is pure, the efficiency would be one (100%). Still, in this case, the battery would be of no use, since the amount of stored energy is very low. Hence, I think that one cannot focus only on the efficiency. As a minor remark, the authors use P(\tau) to denote the efficiency. In the field, this usually denotes the average power, E(\tau)/\tau.
3.  The results obtained with a Lindblad master equation are compared with the ones obtained with a Redfield master equation,  which is claimed to be more accurate. While this is often true, the Lindblad master equation has the property of being Completely Positive and Trace-Preserving (CPTP).  Briefly, this means that all the states produced by the Lindblad master equation are physical. The same does not hold for the Redfield, which is not guaranteed to be CPTP and can yield unphysical results.  The manuscript should address whether the steady states obtained via the Redfield equation are physical and comment on the implications of using non-CPTP dynamics.
4.  The manuscript suggests a link between efficiency and entanglement. This observation appears to be more coincidental and specific to the setup rather than indicative of a deeper principle. The relevance of this finding should be critically examined.
5.  In Eq. 3, the bath's degrees of freedom are given by c_k and b_k.  Are these fermionic and bosonic operators?
The authors also state that "In this paper, we further couple the charger and battery with two independent reservoirs that can exchange energy (for boson reservoirs) or particles (for fermion reservoirs) with the system". Can the authors say more about this particle exchange?
6. In general, the article is written in poor English. There are also a few typos.
The manuscript starts with:
"The stage is yours. Write your article here. The bulk of the paper should be clearly divided into sections with short descriptive titles, including an introduction and a conclusion." which clearly should be removed!
Below,  I guess that "Floquent" should be "Floquet".

---

## Round 2 · Referee Report · Anonymous (Referee 1) · 2024-12-17

Report

In the second version of the manuscript, the authors addressed basically all of the remarks I pointed out in the previous report. While the paper has significantly improved after the revision, I believe there is still perhaps a bit of confusion for what concerns the various approximations involved, as I will elaborate shortly. However, once the following minor points are addressed, I think the paper can be published, even though its current scope could be better suited to SciPost Physics Core instead of SciPost Physics.

- From reading the abstract and the introduction, it seems that treating the charger-battery system as a whole is what drives the need to go beyond the secular approximation towards the Redfield equation. From reading the end of Section 2.1 it seems instead that the Redfield equation is needed in order to treat nonequilibrium reservoirs. I think that both of these impressions are misleading, as nonequilibrium is not directly related to the secular approximation. Specifically, the secular approximation can be applied whenever $\min_{i \neq j} |\epsilon_{ij}|$ is greater than the system-reservoir coupling: this does not exclude the possibility of having nonequilibrium reservoirs. If one wants to consider the charger-battery system as a whole, the problem of Eq. (6) is not the secular approximation but the space locality, i.e. the Lindblad operators are the ladder operators of charger and battery separately. A Lindblad equation can also be derived using a global approach, simply by performing the secular approximation on Eq. (12). It is probably less accurate than the Redfield equation, but it is already capable of accounting for the charger-battery coupling.
- If one starts directly with an excitation-preserving interaction of the form (3), I do not see the need to specify twice [under Eq. (5) and under Eq. (10)] that a rotating wave approximation has been made. Actually, in the case of fermionic environments it is more common to describe system-environment hopping using (3) from the very beginning, and no approximation is involved.
- It is not stated anywhere that the master equation here employed is only valid under the crucial assumption of weak coupling between the qubits and their respective environments (I hope I didn't missed that).
- In the answer to my previous remarks a possible explanation has been provided for the efficiency drop at $\mu \sim 0$. It is also stated that such an explanation does not appear in the manuscript, even though I see it at the beginning of Section 4. By the way, I did not completely understand the argument put forward by the Authors: why inhibited particle exchange between the charger-battery system and the environment should result in zero efficiency? What would happen in an ideal case where the environment is not present?

Finally, a few other suggestions:
- In Section 2 it is now clearly stated what is the unitary transformation used to move to the rotating frame, even though it would be nice to clearly state also how such a transformation is carried out, for completeness. Morever, the same symbol $U_1(t)$ is used for two different quantities, see under Eq. (3) and under Eq. (17).
- In Eqs. (13)-(14) the Authors introduced the symbol $\epsilon_{ij}$ without defining it, in contrast to the previous version of the manuscript.
- As far as I know, the words "eigen values", "eigen energies", and similar should be spelled "eigenvalues", "eigenenergies", etc., as already mentioned in my previous report.
- For what concerns the presentation: the text is now erroneously formatted using a bold font everywhere; figures could be made bigger to facilitate reading the labels; hyperlinks seem to be broken in the PDF I received.

Recommendation

Ask for minor revision

  • validity: -
  • significance: -
  • originality: -
  • clarity: -
  • formatting: -
  • grammar: -

Author:  Zhihai Wang  on 2025-02-14  [id 5220]

(in reply to Report 1 on 2024-12-17)
Category:
answer to question

Comment 0-In the second version of the manuscript, the authors addressed basically all of the remarks I pointed out in the previous report. While the paper has significantly improved after the revision, I believe there is still perhaps a bit of confusion for what concerns the various approximations involved, as I will elaborate shortly. However, once the following minor points are addressed, I think the paper can be published, even though its current scope could be better suited to SciPost Physics Core instead of SciPost Physics.

Reply-We sincerely thank the referee for the positive evaluation of the second version of our manuscript. In the revised version, we have carefully considered the referee's suggestions and have made appropriate modifications to the presentation.

Comment 1- From reading the abstract and the introduction, it seems that treating the charger-battery system as a whole is what drives the need to go beyond the secular approximation towards the Redfield equation. From reading the end of Section 2.1 it seems instead that the Redfield equation is needed in order to treat nonequilibrium reservoirs. I think that both of these impressions are misleading, as nonequilibrium is not directly related to the secular approximation. Specifically, the secular approximation can be applied whenever $min ⁡ϵ_{ij}$ is greater than the system-reservoir coupling: this does not exclude the possibility of having nonequilibrium reservoirs. If one wants to consider the charger-battery system as a whole, the problem of Eq. (6) is not the secular approximation but the space locality, i.e. the Lindblad operators are the ladder operators of charger and battery separately. A Lindblad equation can also be derived using a global approach, simply by performing the secular approximation on Eq. (12). It is probably less accurate than the Redfield equation, but it is already capable of accounting for the charger-battery coupling.

Reply -We apologize for the misleading statement in the original version. We agree with the referee that the global master equation can still be expressed in a Lindblad form within the framework of the secular approximation, even when considering the charger-battery system as a whole. However, in our study, we go beyond the secular approximation for two main reasons. First, the external driving affects the original energy spectrum of the charger-battery system. Second, the system is in non-equilibrium. Regarding the second point, we have demonstrated that non-secular terms vanish in the equilibrium case without external driving, as shown in our previous work (Phys. Rev. A 99, 042320 (2019) and New J. Phys. 20, 033034 (2018)). Therefore, the non-equilibrium nature of the system motivates us to go beyond the secular approximation in this study.

Comment 2 - If one starts directly with an excitation-preserving interaction of the form (3), I do not see the need to specify twice [under Eq. (5) and under Eq. (10)] that a rotating wave approximation has been made. Actually, in the case of fermionic environments it is more common to describe system-environment hopping using (3) from the very beginning, and no approximation is involved.

Reply- The interaction Hamiltonian in our work is an excitation-preserving Hamiltonian, but it is not exactly the Hamiltonian under the rotating wave approximation (RWA). The RWA refers to retaining only the "slow variable terms" in the system-reservoir interaction Hamiltonian in the interaction representation. In the presence of external driving, the excitation-preserving Hamiltonian and the RWA are not strictly equivalent. In fact, we perform the RWA in Eq. (10).

In the revised manuscript, we have removed the related words under Eq. (5), while retaining those under Eq. (10).

Comment 3- It is not stated anywhere that the master equation here employed is only valid under the crucial assumption of weak coupling between the qubits and their respective environments (I hope I didn't missed that).

Reply-We appreciate the referee's kind reminder. In the revised version, we explicitly state before Eq. (11) that the master equation used in this work is valid only under the weak system-reservoir coupling assumption.

Comment 4- In the answer to my previous remarks a possible explanation has been provided for the efficiency drop at μ∼0 It is also stated that such an explanation does not appear in the manuscript, even though I see it at the beginning of Section 4. By the way, I did not completely understand the argument put forward by the Authors: why inhibited particle exchange between the charger-battery system and the environment should result in zero efficiency? What would happen in an ideal case where the environment is not present?

Reply-During the dynamical evolution, the charger-battery system exchanges both particles and energy with the environment to complete the charging process. However, when the chemical potential μ approaches zero (μ ~ 0), the particle exchange is suppressed, leaving only the energy exchange intact. As a result, we observe a sudden drop in charging efficiency. Furthermore, due to the presence of energy exchange, a zero chemical potential does not imply the absence of the environment.

In the absence of the environment, the charger-battery system undergoes unitary evolution, and the efficiency oscillates between 1 and 0 due to the coupling between the charger and the battery. However, it does not reach a steady state.

Finally, a few other suggestions:
Comment 5- In Section 2 it is now clearly stated what is the unitary transformation used to move to the rotating frame, even though it would be nice to clearly state also how such a transformation is carried out, for completeness. Moreover, the same symbol U1(t)U1(t) is used for two different quantities, see under Eq. (3) and under Eq. (17).

Reply- In the revised version, we explicitly clarify how the transformation is carried out in Sec. 2, namely:

$ H^'=U_1 HU_1^++iU_1 ∂/∂t U_1^+$

Furthermore, we emphasize that the transformation in Sec. 2 applies to both the charger-battery system and the reservoirs, whereas the transformation under Eq. (17) is performed only for the charger-battery system, as the reservoirs have already been traced out in the master equation.

Comment 6- In Eqs. (13)-(14) the Authors introduced the symbol$ ϵ_ij$ without defining it, in contrast to the previous version of the manuscript.

Reply-In the revised version, we ensure that all symbols are clearly defined for clarity and consistency.

Comment 7- As far as I know, the words "eigen values", "eigen energies", and similar should be spelled "eigenvalues", "eigenenergies", etc., as already mentioned in my previous report.

Reply-We apologize for the oversight. In the revised version, we carefully review and correct any spelling errors.

Comment 8- For what concerns the presentation: the text is now erroneously formatted using a bold font everywhere; figures could be made bigger to facilitate reading the labels; hyperlinks seem to be broken in the PDF I received.

Reply-We have followed the formatting guidelines provided by SciPost Physics. In the revised version, we enlarge certain figures and their corresponding labels to enhance readability and improve the convenience for readers.

---

## Round 2 · Author Response

Errors in user-supplied markup (flagged; corrections coming soon)

Thank you for informing us of the referee's report on our manuscript (202306-00038v1/Wang). We also extend our gratitude to the referee and editors for their comments and helpful suggestions. We have carefully revised our manuscript by incorporating these comments.

Given the extensive modifications in both the physical content and the English expressions, we have not labeled all individual changes. Instead, we provide a version of the revised manuscript highlighting the major changes in red, and we hope this does not cause any inconvenience to the editors or referees during the second-round review. Please see the detailed point by point answers to the questions raised by the referees below.
* * *
Report of the First Referee--202306-00038v1/Wang
* * *
Comment:

0. This manuscript presents an investigation into an open quantum battery system, consisting of a qubit coupled to a driven qubit (referred to as the charger) and in contact with two thermal reservoirs, both fermionic and bosonic. The study utilizes both Lindblad and Redfield master equations, exploring phenomena such as bistability.

In my opinion, this manuscript fails to do an appropriate analysis due to several important shortcomings. I find this manuscript to be poorly written and I have found important typos. I believe this article could be published in SciPost Physics only after a serious and major revision.

Here are the main points that need addressing.

Reply:

We thank the referee for the excellent summarization of our work. Following the comments and suggestions from both referees, we have made significant modifications throughout the paper, including both the physical content and the English expressions.

The point-to-point reply to all comments is provided as follows:

Comment:

1. I am not persuaded that the authors are studying something that could be called a ``battery". The authors study the steady state of an out-of-equilibrium problem and report the charging efficiency. It is not clear, however, how this system, is continuously driven by the baths and the classical field, could operate as a battery, storing energy for a long time and providing it to a third system to do something useful.

Reply:

We thank the referee for the insightful comments. It is true that the model we study consists of two identical qubits, with one of them being continuously driven. This system can be considered as a minimal model for a quantum battery for the following reasons:

(1). In the charging process, a charger interacts with an external energy source and transfers energy to another part, i.e., the battery. This is precisely the process studied in our current manuscript.

(2). In our revised manuscript, we consider both qubits immersed in fermionic reservoirs. This setup closely resembles the behavior of a real ``battery'' system, even in classical contexts.

(3). We have investigated the charging efficiency and power in our quantum battery setup. The charging process is considered complete when the interaction between the two qubits is turned off at a specified moment, $\tau$. At this point, the charging energy, $E(\tau)$, is stored in the battery qubit. While ergotropy quantifies the energy that can be extracted from the battery, the question of how to use the battery to supply energy to a third party is beyond the scope of our current study. In this sense, our work focuses on the performance of the quantum battery setup rather than its practical usage. Therefore, we believe our study remains an integral part of research on quantum battery preparation and design.

Comment:

2. The manuscript defines efficiency as the ratio between ergotropy and average energy. think it would be much better to show also the average energy, as the efficiency alone is not indicative of the energy injected in the battery. As an example, if the battery has a very low amount of energy, but the battery's state is pure, the efficiency would be one. Still, in this case, the battery would be of no use, since the amount of stored energy is very low. Hence, I think that one cannot focus only on the efficiency. As a minor remark, the authors use $P(\tau)$ to denote the efficiency. In the field, this usually denotes the average power, $E(\tau)/\tau$.

Reply:

We thank the referee for the reminder. In the revised version, we have added a discussion on the charging power, $E(\tau)/\tau$. We find that, similar to the efficiency, the compensation mechanism also plays a significant role in enhancing the charging power of our quantum battery setup.

Additionally, as suggested by the referee, we have adopted the notation $R(\tau)$ to denote the efficiency and $P(\tau)$ for the charging power in the revised manuscript.

Comment:

3. The results obtained with a Lindblad master equation are compared with the ones obtained with a Redfield master equation, which is claimed to be more accurate. While this is often true, the Lindblad master equation has the property of being Completely Positive and Trace-Preserving (CPTP). Briefly, this means that all the states produced by the Lindblad master equation are physical. The same does not hold for the Redfield, which is not guaranteed to be CPTP and can yield unphysical results. The manuscript should address whether the steady states obtained via the Redfield equation are physical and comment on the implications of using non-CPTP dynamics.

Reply:

It is true that the Redfield master equation cannot guarantee the positivity of the density matrix, as we have investigated in our previous work [Phys. Rev. A {\bf 99}, 042320 (2019)]. However, this issue arises only within a very small parameter regime. Fortunately, in our current work, we find that the Redfield master equation works well across the entire considered parameter regime.

In the revised manuscript, we have included a comment on this point in the conclusion section.

Comment:

4. The manuscript suggests a link between efficiency and entanglement. This observation appears to be more coincidental and specific to the setup rather than indicative of a deeper principle. The relevance of this finding should be critically examined.

Reply:

We thank the referee for the kind reminder. Regarding Fig.~2(a) in the revised manuscript, we have also investigated the charging efficiency as a function of the detuning $\Delta/\lambda$, aiming to explore the connection between entanglement and efficiency.

Unfortunately, we can not specify the exact role of entanglement in determining the charging efficiency. By comparing Fig.~2(a) in the revised manuscript and the efficiency (not shown here), we observe that the line shapes and the positions of the maxima for entanglement versus detuning $\Delta/\lambda$ and efficiency versus detuning $\Delta/\lambda$ are different.

As a result, we have not included this additional result in the revised manuscript. Instead, we treat entanglement and charging efficiency as two separate sections.

Comment:

5. In Eq. 3, the bath's degrees of freedom are given by $c_k$ and $b_k$. Are these fermionic and bosonic operators? The authors also state that "In this paper, we further couple the charger and battery with two independent reservoirs that can exchange energy (for boson reservoirs) or particles (for fermion reservoirs) with the system". Can the authors say more about this particle exchange?

Reply:

First, in the revised version, we have discarded the discussion on bosonic reservoirs, as real battery systems are typically connected to fermionic reservoirs. Consequently, $c_k$ and $b_k$ in the revised manuscript are now treated as fermionic operators.

Second, the chemical potential difference between the two reservoirs drives particle exchange until chemical equilibrium is reached. Thus, in our setup, when the two reservoirs are in a nonequilibrium state, they exchange fermions through the charger-battery coupled system.

Comment:

6. In general, the article is written in poor English. There are also a few typos.
The manuscript starts with:
"The stage is yours. Write your article here. The bulk of the paper should be clearly divided into sections with short descriptive titles, including an introduction and a conclusion." which clearly should be removed!
Below, I guess that "Floquent" should be "Floquet".

Reply:

We apologize for our carelessness. In the revised version, we have made every effort to improve our English expression.
* * *
Report of the Second Referee -- 202306-00038v1/Wang
* * *
Comment:

0. In this work the authors study several improvements over the standard scenario of a qubit quantum battery charged by a driven-dissipative qubit ancilla. In particular, they add a different noise on the battery and they treat the whole model using a microscopically sound approach. They derive the corresponding Redfield equation at weak coupling and numerically solve it to study the behavior of the efficiency in various parameter regimes. I really liked how they sticked to a more realistic steady-state charging protocol (instead of the classical switch on/off), and their promising results about how nonequilibrium features can enhance the efficiency in a less traditional nonresonant driving scheme. However, before expressing my final recommendation I would like the authors to address the following remarks.

Reply:

We thank the referee for the positive comments on our work. In particular, we appreciate the referee's recognition of our approach as ``stick to a more realistic steady-state charging protocol'' and the acknowledgment of our ``promising results about how nonequilibrium features can enhance the efficiency in a less traditional nonresonant driving scheme.''

We will address all the comments one by one and revise our manuscript accordingly.

Comment:

1. It is not entirely clear from the presentation how the rotating frame transformation at frequency $\omega_d$ interacts with the rotating wave approximation. Specifically, from Eqs. (1)-(2) I can guess that the starting universe Hamiltonian has been rotated with something like $\exp[i\omega_d(\sigma_z^{(1)}+\sigma_z^{(2)})t/2+i\omega_d\sum_k(b_k^\dagger b_k+c_k^\dagger c_k)t]$, but to me this is not a completely trivial fact and should be specified. This is perhaps partially mentioned below Eq. (25), but a factor $\omega_d/2$ is missing there, and the reservoir part obviously do not appear. If this is the correct transformation, then I believe the expression Eq. (3) for the system-reservoir coupling $V$ is not the rotated one, since counter-rotating terms should present a factor $\exp(2i\omega_d t)$. This does not impact the results of the paper, since the rotating wave approximation is performed from Sec. 2.1 onward: as a consequence, I suggest to remove counter-rotating terms from Eq. (3) and stick with the rotating-wave form throughout, unless I missed some use of counter-rotating terms in the rest of the paper.

Reply:

First, it is true that the Hamiltonian is rotated with
$U_1 = \exp\left[i\omega_d\left(\frac{\sigma_z^{(1)} + \sigma_z^{(2)}}{2}\right)t + i\omega_d\sum_k(b_k^{\dagger} b_k + c_k^{\dagger} c_k)t\right].$
In the revised version, we have not only explicitly presented this transformation but also provided the initial Hamiltonian before the rotation.

Second, we have corrected the expression below Eq.~(25) in the original manuscript, which corresponds to Eq.~(17) in the revised version. Here, the density matrix describes the charger-battery system, with the operators for the reservoirs traced out. Consequently, the reservoir part does not contribute and does not need to be included in this expression.

Third, in the original manuscript, we incorrectly left the interaction Hamiltonian unrotated. As pointed out by the referee, this leads to the presence of counter-rotating terms with a factor of $\exp(2i\omega_d t)$. Following the referee's suggestion, we have now applied the rotating wave approximation. However, we find that this prevents us from obtaining analytical results in the resonantly driven case, and the bistable behavior also disappears. Therefore, we have removed related discussions on analytical results and bistable behaviors in the revised manuscript.

Comment:

2. I do not understand why the constants $K_{\pm}$ are introduced below Eq. (10), since they are identical to the $G_{\pm}$ constants. Since $U^{-1}=U^\dagger$, why not use the same notations of Eq. (9)? Moreover, it should be true that $\omega_\pm/G_{\pm}=K_{\pm}/2M$, but it is not immediately clear to me that such a relation holds.

Reply:

We are sorry but slightly puzzled by the referee's statement.

First, in our manuscript, we have defined $G_{\pm} = \sqrt{F^{2} + \lambda^{2} \pm \lambda\sqrt{F^{2} + \lambda^{2}}}$ and $K_{\pm} = \sqrt{F^{2} \pm 2\lambda\omega_{\pm}}$, so $K_{\pm} \neq G_{\pm}$.

Second, the transformation $U$ is unitary rather than Hermitian, meaning that $U^{-1} = U^{\dagger}$, not $U^{-1} = U$. Therefore, it is not necessary to satisfy the relation $\omega_\pm / G_{\pm} = K_{\pm} / 2M$.

Third, we have removed this analytical part from the revised manuscript due to the reasons mentioned in our reply to the previous comment.

Comment:

3. Eq. (14) shows the general form of the Redfield equation in interaction picture, while Eq. (15) shows its rielaborated form in Schroedinger picture, but the same symbol $\rho$ for the system state is used: I suggest to differentiate the notation to avoid confusion. Moreover, Eq. (14) is written with an interaction $V$ that is $\omega_d$-rotated, but it is not clear if Eq. (15) is rotated as well or not. I guess it is still in the rotated frame, but since in Sec. 2.2 a transformation that inverts the $\omega_d$ rotation is introduced, some clarifications can avoid misunderstanding. I would also add a citation to Breuer-Petruccione just above Eq. (14) and explicitly write that the Redfield equation is obtained under a weak-coupling assumption.

Reply:

First, following the referee's suggestion, we will denote the density matrix in the interaction picture as $\rho^{I}$ in Eq.~(14) and that in the Schroedinger picture as $\rho$ in Eq.~(15). These equations now correspond to Eqs.~(9) and (10) in the revised manuscript.

Second, Eq.~(15) [Eq.~(10) in the revised version] is still formulated in the rotated frame because $|E_i\rangle \,(i=1,2,3,4)$ are the eigenstates of the rotated Hamiltonian.

Third, to clarify, we have cited Breuer-Petruccione's textbook and explicitly pointed out that the approach operates under the weak-coupling assumption.

Comment:

4. A similar notation problem happens in Eq. (28), where the symbol $H_B$ appears to refer to the battery Hamiltonian, even though the same symbol is used in Eq. (2) for the reservoir Hamiltonian. Moreover, it is stated that $H_B=\omega|e\rangle\langle e|$
but with the previous notations it should be $\omega_2/2(|e\rangle\langle e|-|g\rangle\langle g|)$. I believe this energy shift does not influence the results, but uniformization can make the paper more clear.

Reply:

We thank the referee for the reminder. In the revised version, we have clarified that $H_B = \omega_2/2 (|e\rangle\langle e| - |g\rangle\langle g|)$ to ensure consistency with the previous definition. Accordingly, the mean charging energy $E_B(\tau)$ is defined as
$E_B(\tau) = {\rm Tr} [H_B \rho_B(\tau)] - {\rm Tr} [H_B \rho_B(0)].$

Comment:

5. Below Eq. (37) it is stated that the concurrence is used to measure the stead-state entanglement between charger and battery. I would briefly mention the expression you used to calculate it, just to make the paper more self-contained.

Reply:

We thank the referee for the beneficial suggestion. In the revised version, we have added a new paragraph (the second paragraph in Sec.~3) to briefly introduce the expression of the concurrence. Additionally, we have cited Wootters' work as a reference.

Comment:

6. The Conclusion section beautifully summarizes the obtained results, but my feeling is that the same cannot be said about the abstract. The one submitted to arXiv is written considerably better. In both versions, however, it is not clear that the "compensation mechanism" the authors mention is related to having temperature or chemical potential difference: in my view, the efficiency enhancement that nonequilibrium features bring to the nonresonant driving scheme is the most important result of the paper and should be emphasized more in the abstract.

Reply:

In the revised version, we have modified the abstract to clarify the meaning of the ``compensation mechanism.'' Meanwhile, following the referee's suggestion, we have emphasized the role of non-resonance in enhancing the performance of the quantum battery, not only in terms of charging efficiency but also in power. Furthermore, we have revised the conclusion section to improve its clarity and presentation.

Comment:

7. It is interesting to see how assuming resonant driving opens up a nontrivial steady-state space. In Lindblad theory, the closure of the Liouvillian gap is often associated with dissipative phase transitions (see Ref. [58]). I guess some subtleties should be taken into consideration when looking at the Redfield equation, but do you think it is possible to interpret the bistability you obtain using arguments from DPTs and symmetry breaking? A related question is why the steady-state concurrence in Fig. 3 appears to be so discontinuous as a function of $\Delta$.

Reply:

We agree with the referee that bistability is closely related to dissipative phase transitions (DPT) and symmetry breaking. However, as mentioned earlier, our previous submission did not properly account for the rotating frame. After correcting this oversight, we find that bistability no longer appears. Consequently, we are unable to connect our results to DPT, and the steady-state concurrence as a function of $\Delta/\lambda$ becomes continuous, as shown in Fig.~2 of the revised manuscript.

Comment:

8. In Fig. 7b you show how the efficiency at equilibrium with resonant driving drops for a certain value of $\mu$ that is not dependent on $T$. Do you have an interpretation for why this happens and/or why it happens at that specific value of
$\mu$?

Reply:

We acknowledge that we are unable to provide a definitive answer to this comment. However, we observe that this minimum value appears at a position slightly greater than $0$. One possible explanation is that when the chemical potentials of both reservoirs are zero, they cannot exchange particles with the charger-battery system, resulting in low or potentially zero efficiency. The minor deviation might originate from the external driving applied to the charger.

These are speculative interpretations and, therefore, have not been included in the revised manuscript.

Comment:

9. It is a known fact in the theory of open quantum systems that the Redfield equation is not positive, meaning that it could lead to the prediction of meaningless negative probabilities for measurement outcomes. Did you have problems in this sense, or your values for $\alpha_{1,2}$ were sufficiently low to not encounter issues?

Reply:

It is true that the Redfield master equation sometimes fails to guarantee the positivity of the density matrix. Fortunately, in our current work, we have verified the positivity across the entire considered parameter regime. One possible reason for this might be the relatively small values of $\alpha_{1(2)}$, as noted by the referee.

Comment:

10. There is a nonnegligible number of grammatical and orthographic issues. For example: "nonequilibrium" is misspelled several times; "eigen states" and "eigen values" instead of "eigenstates" and "eigenvalues"; other statements are ill-formed, especially in the Introduction (e.g., "the quantum battery which is subject to the open system"). There is also a piece of the SciPost template the beginning of the Introduction. I suggest to perform a complete writing recheck.

Reply:

We apologize for the carelessness in our English expressions in the original manuscript. In the revised version, we have thoroughly polished the language to the best of our ability and hope it now meets the standards of SciPost Physics.

Comment:

11. Related to the previous point, I noticed typos in some mathematical expressions. Below Eq. (4), it should be $\delta(\omega-\omega_{bk})$ instead of $\delta(\omega-\omega_1)$ and $\delta(\omega-\omega_{ck})$ instead of $\delta(\omega-\omega_2)$. In Eq. (8), $E_3$ is repeated twice, whereas
$E_4$ should appear instead in second occurrences. Below Eq. (25), $\omega_d/2$
should appear in $U_1(\tau)$ I believe.

Reply:

We thank the referee for the kind reminder. In the revised manuscript, we have corrected the above typos.

Comment:

12. Few suggestions about the figures: in Fig. 2 the label $\rm {R}[\Lambda/\lambda]$
is used to indicate the Liouvillian gap but in the text the notation $\Lambda={\rm R}[\lambda_1]$ is used; in Fig. 3 maybe it should be specified in the caption that the tomography is performed at $\delta=0$; in Fig. 4 the majority of the information is concentrated around $\delta=0$ but the plot is too zoomed out to appreciate the chemical potential differences; Figs. 9-11 are maybe too small.

Reply:

Due to the disappearance of the bistable behavior, we have removed Figs.~2 and 3 from the revised manuscript. Meanwhile, we have enlarged Figs.~9-11, leaving only one figure, which is now presented as Fig.~4. Additionally, we have standardized all the figures in the manuscript.

We sincerely thank the referee and editor for their helpful comments and suggestions. We have revised the manuscript accordingly and believe that these modifications have significantly improved its quality. We hope that the revised manuscript meets the standards and can be accepted for publication in SciPost Physics.

Sincerely,

Zhihai Wang, Hongwei Yu and Jin Wang

---

## Round 2 · List of Changes

1. We have corrected the errors in the rotating frame transformation.
2. A discussion on the charging power of the quantum battery has been added, by immersing the charger-battery system in Fermionic reservoirs. This investigation reinforces its designation as a "battery" .
3. A brief introduction to the definition of concurrence has been included.
4. The abstract, introduction, and conclusion sections have been revised to further emphasize our key findings.
5. Errors and typos throughout the manuscript have been corrected.
6. New references have been incorporated to enhance the context and depth of the study.

---

## Editorial Decision

in_refereeing